# Immunogenicity Characterization of the Recombinant gI Protein Fragment from Pseudorabies Virus and an Evaluation of Its Diagnostic Use in Pigs

**DOI:** 10.3390/vetsci10080506

**Published:** 2023-08-05

**Authors:** Haijuan He, Baojie Qi, Yongbo Yang, Xiaowen Cui, Tianfeng Chen, Xuehui Cai, Tongqing An, Shujie Wang

**Affiliations:** 1National Key Laboratory for Animal Disease Control and Prevention, Harbin Veterinary Research Institute, Chinese Academy of Agricultural Sciences, Harbin 150068, China; he_haijuan@haas.cn (H.H.); xqj111666777@163.com (B.Q.); yangyongbo@caas.cn (Y.Y.); chentf3176@163.com (T.C.); 2Institute of Animal Husbandry, Heilongjiang Academy of Agriculture Sciences, Harbin 150086, China; 3Heilongjiang Minzu College, Harbin 150066, China; 18646310061@163.com; 4Heilongjiang Research Center for Veterinary Biopharmaceutical Technology, Harbin 150068, China; 5Heilongjiang Provincial Key Laboratory of Veterinary Immunology, Harbin 150068, China

**Keywords:** pseudorabies virus, envelope glycoprotein I, immunogenicity, Ni-NTA chromatography, ELISA

## Abstract

**Simple Summary:**

Pseudorabies virus (PRV) is a double-stranded linear DNA virus with an envelope, and it can cause acute neurological symptoms and death in piglets, which has caused large economic loss in pig farms. The eradication of PRV in pig farms has always been a goal, and the diagnosis of wild virus infection is particularly important in this process. In this study, we designed specific primers to amplify the fragment of gI with the region of antigen epitopes and the transmembrane regions removed. The recombinant truncated fragment of gI was expressed, and the native gI protein was obtained. After a small-scale ELISA test for clinical serum samples, it was found that purified gI protein may potentially be applied to clinically diagnose PRV infection.

**Abstract:**

Serological testing is an important method for the diagnosis of pseudorabies virus (PRV) infection. We aimed to investigate the envelope glycoprotein I (gI) of PRV, a strong immunogen, and its potential as an efficient and low-cost diagnostic reagent. In this study, the DNA of the PRV SC strain was used as the template, and the recombinant fragment of gI (633 bp) was amplified via PCR using synthetic primers, and was then ligated into the pET-30a expression vector. The constructs were transferred into *Escherichia coli* (*E. coli*) for prokaryotic expression, and the antigenicity of the expression products was identified by Western blot analysis with pig positive serum against PRV. The recombinant protein was purified by a Ni column, and BALB/c mice were immunized with purified gI protein to obtain anti-gI-positive serum. After PK-15 cells had been infected by PRV for 48 h, the immunogenicity of purified gI protein was identified with a fluorescence immunoassay using anti-gI mouse serum. The recombinant plasmid (pET-30a-gI) was expressed, and the native gI protein was obtained after denaturation by urea and renaturation by dialysis. A small-scale ELISA test containing 1.0 µg/mL of purified gI protein was designed to evaluate pig serum (80 samples), and the results of the ELISA test were compared to those of competitive ELISA (cELISA) tests using IDEXX Kits, which resulted in 97.5% consistency. The results suggested that the truncated gI protein may be a potential diagnostic reagent.

## 1. Introduction

Pseudorabies, also known as Aujeszky’s disease [1,2], is one of the most severe fulminant diseases in livestock, and is caused by the pathogen pseudorabies virus (PRV) [3]. PRV, an enveloped linear dsDNA virus, is included in the genus *Varicellovirus* of the family *Herpesviridae* [4]. PRV may infect numerous mammals, such as swine, wild boars [5], sheep, goats, cattle [6,7], minks, foxes [8] and bears [9]. In particular, some reports have shown that PRV may also overcome the species barrier to infect humans [10,11]. Although there are many hosts for PRV, swine are the most significant natural hosts [12]. PRV infection may affect the central nervous system and lead to death in piglets, and the clinical symptoms of sows are nervous tension, motor incoordination and abortion [13].

The length of the PRV genome is approximately 143 kb, which encodes more than 72 proteins [14]. The PRV capsid has a triangulation number of 16 and is composed of 150 hexon major capsid proteins (MCPs), 150 small capsid proteins (SCPs), 11 penton MCPs and triplexes. The twelfth vertex of the capsid is presumed to be a portal, where the PRV packages and releases the genome [15]. The PRV envelope includes essential glycosylation-modified membrane proteins B (gB), gD, gH and gL, also known as nonessential glycoproteins gC, gE, gG, gI, gK, gM and gN. The glycoprotein gE is an important virulent factor, while glycoprotein gI is nonessential during virus replication, but the complex of gE and gI is necessary for the efficient transport of viral particles in neurons [16].

Currently, the main commercial live-attenuated PRV vaccine strains used in China are the Bartha-K61 and HB-98 strains. Many scholars are also developing novel live-attenuated vaccines for different circulating strains. For example, the PRV strain NY in 2012 was used as the parent strain to construct the recombinant virus of gE/gI/TK deletion and was safe for mice [17]. In addition, a circulating variant strain, HeB12, was also used as the parent strain to generate a PRV vaccine strain with gE/gI/TK deletion through serial passage in vitro, and vaccine efficacy tests showed that the novel vaccine strain may provide effective protection against challenges of classical RA and variant TJ12 strains in vivo [18].

Owing to PRV vaccines of gE gene deletion being widely used in China, some serological tests based on gE antibodies of PRV were developed for differentiating wild virus-infected pigs from vaccinated pigs, while gB antibodies were used to evaluate the levels of vaccine-induced immunity. A variety of serological methods, including enzyme-linked immunosorbent assays (ELISAs) [19], serum neutralization tests, and direct immunofluorescence methods, have also been applied in the past 20 years. However, the most widely used approach is ELISA, which has simpler properties and a higher sensitivity. Among different ELISA methods, competitive ELISAs (cELISA) for gB and gI antibodies are applied widely [20,21,22,23], whereas the kits for cELISA targeting the gI antibody based on serological tests in China are expensive.

Here, we cloned and expressed the gI gene and purified PRV gI protein fragments, and the immunogenicity of the purified protein was identified. The ELISA method established by us was used to detect PRV antibodies in 80 clinical serum samples, and the results were compared with those of cELISA test using IDEXX Laboratories.

## 2. Materials and Methods

### 2.1. Virus Strain, Plasmids, Bacterial Strains and Antibodies

The PRV SC strain, porcine polyclonal anti-PRV positive serum and negative serum used in this study were stored in our lab. The pBluescript and pET-30a (+) vectors, *Escherichia coli* (*E. coli*) strains DH_5α_ and BL_21_ (DE3), Luria–Bertani (LB) medium, X-Gal, and isopropyl β-d-thiogalactoside (IPTG) used in this study were purchased from Tiangen, Beijing, China. Foetal calf serum and RPMI 1640 culture medium were purchased from Gibco, USA. Restrictive enzymes *EcoR* V, *Bam* I and *Sal* I and antibiotics were purchased from Dalian Bao Biological Engineering Co., Ltd. (Dalian, China).

### 2.2. Primer Design and Synthesis

According to the protein sequence of PRV envelope, glycoprotein gI (GenBank accession No. AFI70843.1) was used to analyze possible epitopes (Figure 1A) and transmembrane regions of the gI protein (Figure 1B) using Protean software (Madison, WI, USA) and BepiPred/TMHMM-2.0 online server. After removing the transmembrane region, the upstream and downstream primers of PCR were designed using Oligo 7.0 software, and the relative restriction endonuclease sites *Bam* I and *Sal* I were added to upstream and downstream primers, respectively, in order to construct the recombinant clone plasmid of gI. The primers were synthesized by Boshi Inc. (Harbin, China), and the sequences were as follows: primer P1: 5′-TATGGATCCGACGGGACGCTGCTGTTTCT-3′ and primer P2: 5′-AAAGTCGACTTAGAGCAGGACGCGCGACACGACG.

### 2.3. Construction of Recombinant Expression Vector

The genomic DNA of the PRV SC strain was extracted using a Virus DNA Extraction Kit (Omega, Shanghai, China) according to the manufacturer’s manual. The PCR products were loaded onto 1.2% agarose gels and analyzed in the gel image analysis system (ChampGel 5000 Plus, SINSAGE, Beijing, China). The PCR products were purified using a Gel Extraction Kit (Tiangen, China). The purified PCR products were then ligated into the *EcoR* V digested pBluescript vector (Youbao, Shenzhen, China) and transferred into *E. coli* DH_5α_, and positive *E. coli* cell clones were screened on Luria–Bertani (LB) agar plates (Gibico, BRL, Waltham, MA, USA) with 100 μg/mL ampicillin, 30 μg/mL X-Gal and 1 mM/mL IPTG. Finally, positive clones were sent for sequencing by Comate Bioscience Company Ltd.

The corrected plasmid pBluescript-gI was double digested, and the target fragment (633 bp) was ligated to the prokaryotic expression vector pET-30 (a) with T4 DNA ligase at 22 °C for 2 h and transferred into *E. coli* DH_5α_. The recombinant plasmid was identified, and the positive clone was named pET-30-gI. Finally, the recombinant expression plasmid pET-30-gI was transformed into *E. coli* BL_21_ (DE3) competent cells.

### 2.4. Expression and Purification of Truncated gI Protein

The truncated gI protein of PRV was expressed in *E. coli* BL_21_. Positive colonies of *E. coli* BL_21_ cells were selected and cultured in LB broth with kanamycin (100 μg/mL) at 37 °C and the culture was shaken to 0.5 for optical density (OD) at 600 nm. In order to induce the truncated gI protein expression, IPTG with a final concentration of 0.3–1.0 mM/mL was added to the culture and incubated for 7 h. Cultures containing an empty vector pET-30 (a) were also included as a control in parallel. The protean of DNAStar software was used to predict the secondary structure characteristics of expressed truncated proteins.

Bacterial cells were harvested by centrifuging at 7000 rpm at 4 °C for 10 min. Bacterial cells were suspended with buffer I (20 mM Tris-HCl, 5 mM imidazole, 5 mM β-mercaptoethanol, 500 mM NaCl, pH 7.8). The ultrasonic destruction of the collected cell lysate was carried out on ice for 12 min (pulse 5 s, pause 5 s, 200 w). Dissolved cells were centrifuged at 11,000 rpm at 4 °C for 20 min, then suspended with buffer II (Buffer I + 1% Triton X-100).

The inclusion bodies were dissolved with 40 mL buffer solution III (buffer solution I+8 M urea) overnight at 4 °C. Ultrasonic destruction was carried out on ice for 5 min (pulse 5 s, pause 5 s, 200 w). To remove cell debris, the cell lysate was centrifuged at 15,000 rpm for 30 min, and a 0.22 filter membrane was used to filter the supernatant. Then, the filtrate was placed on the Ni histidine binding resin column, which was preequilibrated with binding buffer. The resin was washed extensively with washing buffer III. The urea was dissolved in buffer IV (20 mM Tris HCl, 5 mM imidazole, 1 mM GSH, 500 mM NaCl, 20% glycerol, pH 7.8), then used for renaturation with concentrations ranging from 0 M to 8 M urea.

### 2.5. SDS–PAGE and Western Blot

Immunoblotting was performed as described in the following. Briefly, expressed gI protein products were loaded on SDS-PAGE gels and the gels transferred to a nitrocellulose membrane. The membranes were blocked overnight at 4 °C using 10% dried milk dissolved in phosphate-buffered saline (PBS), and incubated with porcine polyclonal anti-PRV positive serum or negative serum for 1 h at room temperature. After washing in PBS containing 0.05% Tween 20 (PBST), the membrane was incubated with HRP-conjugated goat anti-porcine IgG (1:10,000, Sigma, Saint Louis, MO, USA) for 1 h at room temperature. Immunoreactive bands were visualized using a DAB Substrate Kit for 15 min (thermos scientific, Waltham, MA, USA).

### 2.6. Animal Ethic Statements

This study was carried out according to the Guide for the Care and Use of Laboratory Animals of the Ministry of Science and Technology of China. The Committee on the Ethics of Animal Experiments in Harbin Veterinary Research Institute of Chinese Academy of Agricultural Sciences reviewed and approved the protocols. Purified gI protein was injected into 42-day-old BALB/c mice intraperitoneally and intramuscularly in the animal biosafety level 2 facilities in Harbin Veterinary Research Institute, Chinese Academy of Agricultural Sciences (approval number SY-2015-MI-068).

### 2.7. Preparation of Anti-gI Protein Mouse Serum

To prepare specific anti-gI protein polyclonal antiserum, each 42-day-old female BALB/c mouse was immunized with intraperitoneal and intramuscular injections of 50 μg (0.3 mL) purified gI protein with Fredrick’s complete adjuvant. After that, mice were immunized twice with Freund’s incomplete adjuvant, once every two weeks. Four days after the third immunization, antiserum was collected from the blood of the immunized mice.

### 2.8. Characterization of Murine gI Protein Polyclonal Antibody against PRV

PK15 cells were infected with wild type PRV at a multiplicity of infection (MOI) of 9 and incubated in a 12-well plate for 48 h at 37 °C. Subsequently, the cells in the 12-well plate were fixed with 75% ethanol for 20 min and washed with PBS three times. After that, the murine polyclonal antibody (1:100) against gI protein was used as a primary antibody for 1.5 h at room temperature, and Alexa Fluor^TM^ 488-conjugated goat anti-mouse antibody (1:1000, Sigma, Saint Louis, MO, USA) was used as a second antibody for 1.0 h at room temperature. After washing three times with PBS, the cells were observed under a fluorescence microscope.

### 2.9. Indirect ELISA

Eight pig positive and eight pig negative serum samples that had previously been tested were used for the optimization of the new ELISA based on purified gI protein. In brief, ELISA plates were coated with purified gI protein at different doses (0.05 μg, 0.1 μg, 0.2 μg, 0.3 μg) in carbonate bicarbonate buffer (15 mM Na_2_CO_3_, 35 mM NaHCO_3_; pH 9.6), 100 μL/well. The plates were incubated at 4 °C overnight and washed five times using PBST. Then, 10% milk blocking solution was added to each well of the plate and incubated for 2 h at 37 °C and washed five times using PBST. After washing, pig sera diluted in 5% milk were added (1:50, 1:100, 1:200; 100 μL/well), and plates were incubated for 1 h at 37 °C. After washing the plates five times using PBST, the HRP-conjugated goat anti-pig IgG antibody (Sigma, USA) was diluted (1:5000, 1:8000, 1:10,000) and added to the plate (100 μL/well) and incubated for 1 h at 37 °C. After four washings with PBST, 100 μL of tetramethylbenzidine (TMB) chromogenic solution was added to each well, and the wells were incubated for 15 min at room temperature. Then, 50 μL of stop buffer (2 M HCL) was added, and the OD value was tested at 450 nm. In total, 80 serum samples (from pigs at different farms in different areas; 40 negative and 40 positive) tested using the PRV gpI antibody Test Kit (IDEXX, Westbrook, ME, USA) were used for the preliminary ELISA validation.

## 3. Results

### 3.1. Construction of the Expression Vector

The genomic DNA of the PRV SC strain was used as a template for PCR amplification using specific gI primers. After PCR amplification, a target fragment of approximately 633 bp was obtained (Figure 2A). The PCR fragment was ligated into the pBluscript vector digested with *EcoR* V, and the recombinant plasmid pBluscript-gI was obtained (Figure 2B). Plasmid pBluscript-gI was sent for sequencing, and the sequence results were compared. The results showed that the target fragment was completely consistent with the sequence of the gI gene published in GenBank. Then, the gI fragment was ligated into the digested (*Sal* I and *BamH* I) expression vector pET-30a. The recombinant expression plasmid pET-30-gI was obtained (Figure 2C).

### 3.2. Expression and Identification of Truncated gI Protein

To express the protein smoothly, a truncated recombinant protein spanning the middle 211 aa of the PRV gI protein was generated, which was expressed in *E. coli* after 0.5 and 1.0 mM IPTG induction. SDS-PAGE gel results showed an approximately 38 kDa band expressed in the lysate of bacteria products (Figure 3A). The results of the Western blot suggested that the expressed gI protein was recognized specifically via PRV-positive pig serum (Figure 3B, the original western blot picture is presented in the Appendix A). The results showed that the truncated recombinant gI fragments were correctly expressed. The secondary structure of the truncated gI protein was predicted by Protean software and is shown in Figure 3C.

### 3.3. Purification of Truncated gI Protein

The data in Figure 4A illustrate that the gI protein was expressed mainly in the inclusion body form in *E. coli* after induction using IPTG. After expression, the purification of the gI protein was performed. The inclusion bodies which included gI target protein were collected with ultrasound and centrifugation. The recombinant protein was fully denatured by urea and eluted with nickel affinity chromatography. In order to enable the denatured protein to recover the fold effectively, purified gI protein (Figure 4B) was dialyzed by gradually decreasing the concentration of urea. The recovery rate of the gI protein was 90% after dialysis against different urea concentrations.

### 3.4. Immunogenicity Identification of Truncated gI Protein

The mouse serum used against PRV-truncated gI protein was prepared and the specificity of the polyclonal antibody generated from mice immunized with the truncated gI protein was confirmed with an indirect immunofluorescence assay (IFA). PK-15 cells infected with PRV were detected with the IFA to observe whether the mouse polyclonal antibody could recognize wild-type PRV. Pig anti-PRV-positive serum was used as the positive control, and negative mouse serum and PBS were treated in parallel as the negative control. The results suggested that the mouse polyclonal antibody could react with the PK-15 cells infected with PRV, whereas no fluorescence was found in PBS or the mouse negative serum (Figure 5), which suggested that the truncated gI protein had suitable immunogenicity.

### 3.5. Preliminary Clinical Application of Expressed gI Protein in ELISA

In order to optimize the new ELISA assay based on the expressed gI protein, eight PRV-positive and eight PRV-negative sera were used for gI antibody detection via ELISA. The gI protein was used as the coated protein, and the final coating amount was 0.1 μg/well. The optimal dilution of the tested serum was 1:100. Based on the optimal dilution ratio of coating protein gI and the tested sera, the optimal dilution ratio of the HRP-label goat anti-pig IgG was 1:8000. As shown in Figure 6, under optimized conditions, 40 tested samples coming from the 40 negative sera were negative, and the range of OD value for 40 negative sera samples was 0.067 to 0.209. Thirty-eight samples coming from the 40 positive sera were positive, and the range of OD value for 38 positive sera samples was 0.55 to 2.0 for IgG-specific antibody determinations. The ELISA results were 97.5%, consistent with those of the IDEXX assay.

## 4. Discussion

Pig pseudorabies is a severe infectious disease caused by PRV [24,25], with fever and encephalomyelitis as the main clinical symptoms, which causes huge economic losses to pig farms worldwide [26]. Furthermore, the emerging PRV variants are more pathogenic to pigs than previously isolated strains [27]. The presence of gI antibodies indicates that pigs have been infected with a wild strain of PRV or vaccinated with a vaccine containing the gI protein antigen. Vaccines of gI-deficient PRV are now used on most farms in China, and the viruses used in these vaccines have been selected to eliminate the synthesis of virulence factors, which do not affect the antigenicity of the virus. Therefore, the use of this vaccine enables serological methods to distinguish immune pigs from naturally infected pigs.

In this study, a recombinant gI truncated protein of PRV was expressed in *E. coli*, its immunogenicity was identified, and its potential in establishing PRV-specific IgG antibody ELISA assay was explored. There are four transmembrane helices from inside to outside, including 11–34, 29–51, 152–169 and 285–307 in the full-length (366 aa) gI protein by prediction software. First, we only removed the signal peptide in the C-terminal sequence of gI and retained the N-terminal fragments (921 bp), resulting in no expression of the gI protein (data not shown). The designed gI protein was successfully expressed in *E. coli* in soluble and inclusion body forms after the N-terminal transmembrane region was removed on the basis of removing C-terminal transmembrane helices. This finding suggested that there were more hydrophobic amino acids in the N-terminal transmembrane and that the transmembrane helices were continuous, which was not conducive to protein expression. In a study by W Fuchs et al. [28], recombinant gI proteins (N-terminal sequence, residues 33 to 100) expressed in *E. coli* induced neutralizing and protective antibodies. In this study, our results proved that our expressed protein (residues approximately 60 to 270) had suitable immunogenicity and reactivity.

Since the gI protein we designed is predominantly expressed in inclusion body form, we purified the gI protein using urea denaturation and eluted it via nickel affinity chromatography. After purification, it is important to determine whether the protein can return to its natural conformation. The efficiency of refolding to the original native structure can be increased by increasing the dialyzing time and decreasing the denaturant concentration [29,30]. Therefore, we adopted the method of gradually reducing the concentration of the denaturing solution for protein renaturation. Serum obtained from mice immunized with purified gI protein reacted well with wild type PRV, and IFA showed that the purified gI protein returned to the native structure after denaturation. In addition, there was little gI protein existing in soluble form. We will purify the soluble gI protein and evaluate its reactivity compared with the refolded protein in the future, and decide whether it will be much better than the refolded gI protein for gI-antibody detection in pigs.

We established an indirect ELISA method for the detection of the PRV antibody using purified gI protein as the coated antigen, which tested the negative and positive serum that was detected by the IDEXX assay kit, and the consistency was 97.5%. However, the indirect ELISA method established with the recombinant gI protein may produce a non-specific reaction-related protein of *E. coli*, and will result in false positive results in the actual clinical tests. Therefore, for the coated antigen, it is necessary to purify the protein several times to make it as pure as possible to eliminate the interference of *E. coli* protein. After that, cutting His tags can also reduce associated protein interference. In addition, monoclonal antibodies against the gI protein of PRV should be prepared with purified gI protein, and then blocking ELISA or cELISA should be established to better reduce the non-specific responses.

In conclusion, we developed an ELISA and preliminary application to detect the clinical PRV antibodies, which were 97.5% consistent with those of the IDEXX assay. Our results showed that this approach may potentially be used to clinically diagnose PRV wild strains.

## 5. Conclusions

In summary, we successfully expressed and purified the recombinant gI truncated protein of PRV in *E. coli* and demonstrated its potential as a diagnostic assay for the development of serum tests for PRV infection. The data demonstrate the truncated gI protein as a serological ELISA detection antigen, which may provide a cost-effective reagent for the detection of the PRV antibody, and could provide a method for distinguishing between vaccine-immunized pigs and PRV-infected pigs.

## Figures and Tables

**Figure 1 vetsci-10-00506-f001:**
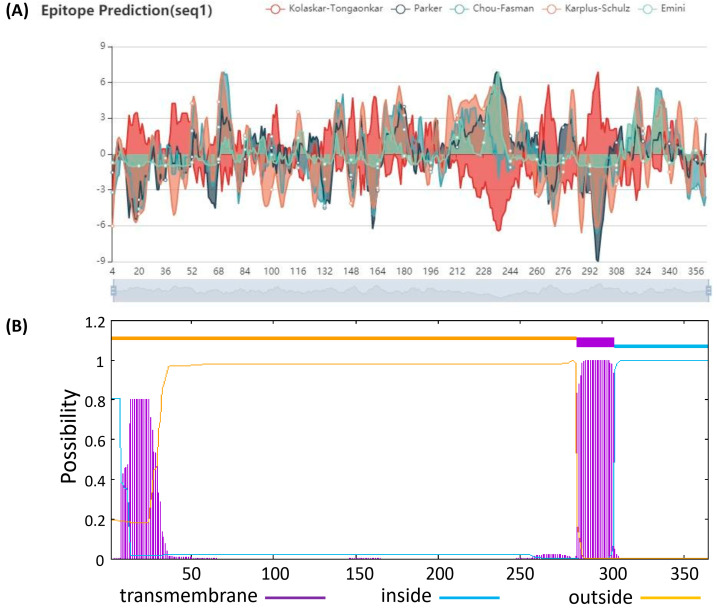
Prediction of possible epitopes and transmembrane regions for gI squence of PRV. (**A**) Possible epitopes for gI squence were predicted by BepiPred online server. (**B**) Possible transmembrane regions for gI squence were predicted by TMHMM-2.0 online server.

**Figure 2 vetsci-10-00506-f002:**
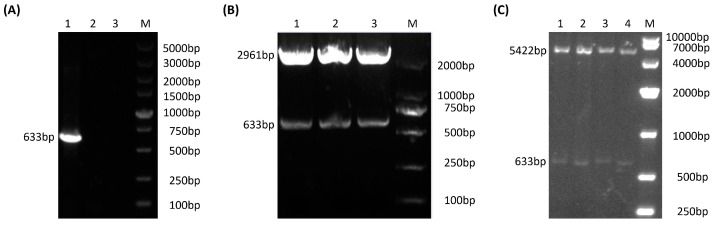
Electrophoresis analysis of PCR products and recombinant plasmid identification of the PRV-gI gene. (**A**) Electrophoretic analysis of PCR products of the PRV-gI gene. Lane 1: PCR product of the gI gene, a target fragment of approximately 633 bp; Lanes 2 and 3: control group; Lane 4: DNA molecular quality standards. (**B**) Recombinant plasmid pBlue-gI enzyme digestion identification. Lanes 1, 2 and 3: *Bam* I and *Sal* I double enzyme cleavage product of pBlue-gI; Lane 4: DNA molecular quality standards. (**C**) Recombinant plasmid pET-30(a)-gI enzyme digestion identification. Lanes 1, 2, 3, 4: Strip analysis using repeated enzyme digestion electrophoresis; Lane 5: DNA molecular quality standards.

**Figure 3 vetsci-10-00506-f003:**
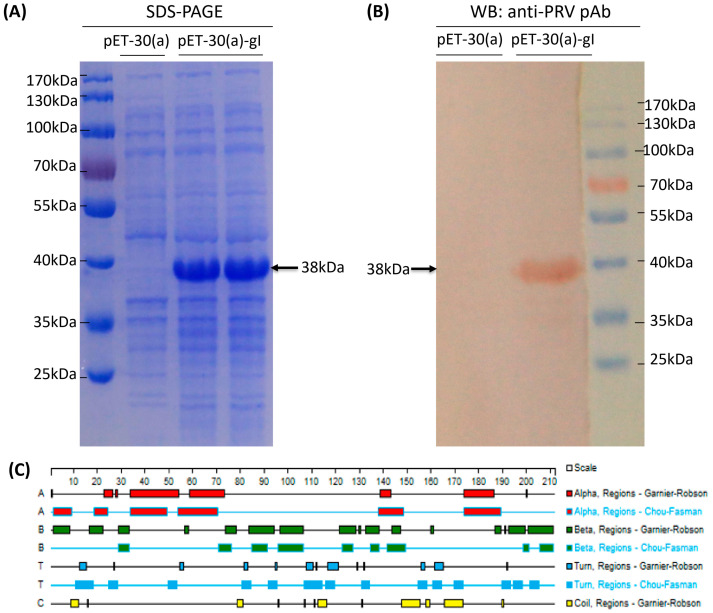
Expression and identification of truncated gI protein. (**A**) SDS-PAGE gel results of target proteins expressed in *E. coli* after 0.3 and 1.0 mM IPTG induction for 7 h at 37 °C. pET-30(a): inducible expression band of empty vector; pET-30(a)-gI: the induced expression of an approximately 38 kDa band of recombinant expression plasmid. (**B**) Western blot analysis was conducted with pig standard positive serum against PRV as the primary antibody; pET-30(a): inducible expression band of empty vector; pET-30(a)-gI: pig standard positive serum against PRV as the primary antibody. (**C**) predicted secondary structure of the monomeric form of the truncated gI peptides generated using Protean software.

**Figure 4 vetsci-10-00506-f004:**
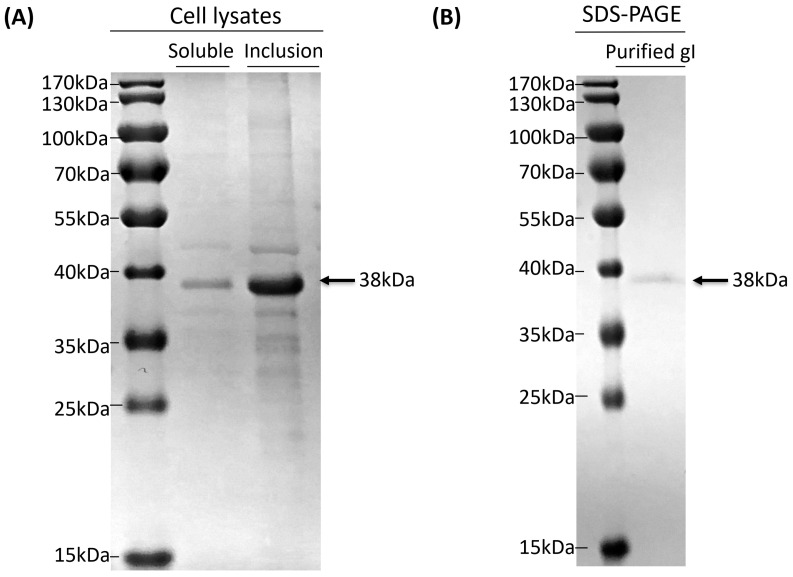
Analysis and purification of the location of the target protein. (**A**) Analysis of target protein supernatant and precipitate via SDS-PAGE gels. Soluble: solubility is expressed in supernatant; inclusion: a protein in the form of an inclusion body. (**B**) Protein purified via nickel column. Purified: purified gI proteins.

**Figure 5 vetsci-10-00506-f005:**
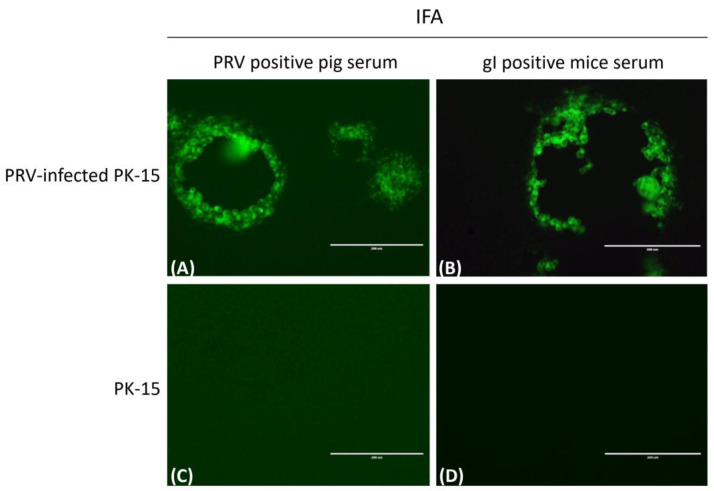
Identification of mouse pAb reacting with PRV via indirect immunofluorescence. PRV infected PK-15 cells for 48 h, and pig anti-PRV-positive serum (**A**) and mouse anti-gI-positive serum (**B**) were used as primary antibodies to detect PRV via IFA; (**C**,**D**) were uninfected PK-15 cells as control, and scale bars represent 200 μm.

**Figure 6 vetsci-10-00506-f006:**
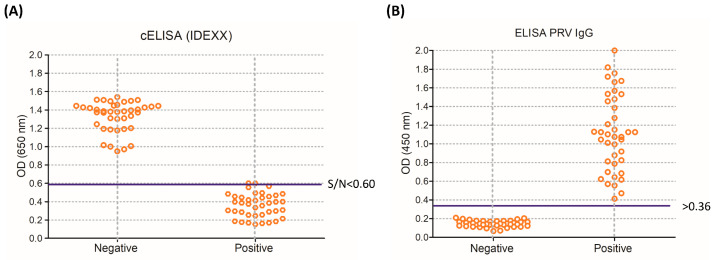
Distribution of the OD values obtained from clinical pig serum samples. Forty PRV-negative and forty PRV-positive pig serum samples using PRV-based IDEXX cELISA (**A**), S/N = Sample A (650)/controls, and (**B**) using gI indirect ELISA.

## Data Availability

All raw data are available upon request.

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
