# Peer review of "Immunogenicity Characterization of the Recombinant gI Protein Fragment from Pseudorabies Virus and an Evaluation of Its Diagnostic Use in Pigs"

_vetsci, 2023, doi:10.3390/vetsci10080506_

Round 1
Reviewer 1 Report
Minor Comment.
1) pseudorabies is also known as Aujeszky's disease, mentioned no where else except the references.
2) When discussing the structure of pseudorabies, this following paper could be mentioned just to give an overall background about pseudorabies structure:
Structures of pseudorabies virus capsids | Nature Communications
3) Is it possible to use such tool/detection method mentioned in this article could be also applicable to Herpesviridae?
Reviewer 2 Report
This manuscript expressed the recombinant gI protein of pseudorabies virus (PRV) in E. coli and evaluated its potential use for detecting gI antibody in an indirect ELISA assay. It would provide some basic data for developing gI-based indirect ELISA used for diagnosis of PRV infection. However, there are some major concerns to be considered before accept for publication.
Major concerns:
1. The gI protein was expressed in E. coli mainly in the form of inclusion bodies, which would decrease significantly its reactivity with the gI-specific antibody. The authors purified the urea denatured gI protein and recovered it by dialysis against different urea concentration. However, the authors did not provide the refolding data such as the recovery rate and antibody reactivity compared with the nature and denatured protein. As shown in Fig. 4B, the recovery rate or purification efficiency might be not so well. The authors should provide more data for purification and refolding the gI expressed protein.
2. As shown in Fig. 4A, there was a little gI protein existed in soluble form. Therefore, the authors are suggested to purify the soluble gI protein and evaluate its reactivity compared with the refolded protein. Maybe it would be much better than the refolded gI protein for gI-antibody detection in pigs.
3. The major concern for the E.coli expressed recombinant protein for antibody detection in an indirect ELISA is the nonspecific reaction related the E. coli protein as well as its less reactivity compared with the nature protein especially for the glycoprotein. Therefore, the authors are suggested to discuss this concern in the discussion section.
Minor concerns:
4. In Fig. 3 and 4, the annotations in the figures should be consistent with the figure legends. i. e. Lanes 1, 2, 3, were not shown in the figures.
5. There are so many typing errors in the manuscript. The authors should recheck them carefully.
Line 28, 193: 633 kb or 633 bp?
Line 35: PET-30a-gI should be pET-30a-gI.
Line 198: Sal I and BamH I should be written in italic.
Line 206: Bam I?
Round 2
Reviewer 2 Report
OK.